# The Adenosine System at the Crossroads of Intestinal Inflammation and Neoplasia

**DOI:** 10.3390/ijms21145089

**Published:** 2020-07-18

**Authors:** Vanessa D’Antongiovanni, Matteo Fornai, Carolina Pellegrini, Laura Benvenuti, Corrado Blandizzi, Luca Antonioli

**Affiliations:** 1Department of Clinical and Experimental Medicine, University of Pisa, Via Roma 55, 56126 Pisa, Italy; v.dantongiovanni@gmail.com (V.D.); laura.benvenuti962@gmail.com (L.B.); c.blandizzi@gmail.com (C.B.); lucaant@gmail.com (L.A.); 2Department of Pharmacy, University of Pisa, Via Bonanno Pisano 6, 56126 Pisa, Italy; carolina.pellegrini87@gmail.com

**Keywords:** adenosine, adenosine receptors, inflammatory bowel diseases, colitis-associated cancer, colorectal cancer, dextran sulfate sodium (DSS)-induced colitis, immune cells

## Abstract

Adenosine is a purine nucleoside, resulting from the degradation of adenosine triphosphate (ATP). Under adverse conditions, including hypoxia, ischemia, inflammation, or cancer, the extracellular levels of adenosine increase significantly. Once released, adenosine activates cellular signaling pathways through the engagement of the four known G-protein-coupled receptors, adenosine A_1_ receptor subtype (A_1_), A_2A_, A_2B_, and A_3_. These receptors, expressed virtually on all immune cells, mitigate all aspects of immune/inflammatory responses. These immunosuppressive effects contribute to blunt the exuberant inflammatory responses, shielding cells, and tissues from an excessive immune response and immune-mediated damage. However, a prolonged persistence of increased adenosine concentrations can be deleterious, participating in the creation of an immunosuppressed niche, ideal for neoplasia onset and development. Based on this evidence, the present review has been conceived to provide a comprehensive and critical overview of the involvement of adenosine system in shaping the molecular mechanisms underlying the enteric chronic inflammation and in promoting the generation of an immunosuppressive niche useful for the colorectal tumorigenesis.

## 1. Introduction

The inflammatory process is a protective response aimed at stemming external insult (i.e., pathogens, toxic compounds, and irradiation) to preserve tissue integrity [1]. However, when inflammation goes beyond this protective purpose, and due to altered molecular mechanisms, there is a chronicization of the inflammatory process, thus creating the conditions for the development of microniches favorable for the development of neoplasia [1].

In this regard, a number of studies well described a causal link between the presence of a chronic inflammatory process and cancer development [2,3,4,5,6]. Indeed, it has been observed that about 20% of neoplastic diseases are observed in patients with a history of chronic inflammatory diseases [2,3,4,5,6]. In this regard, epidemiological investigations pointed out that patients with ulcerative colitis (UC) and Crohn’s disease (CD) are 3–6 times more likely to develop colorectal cancer (CRC) than the general population [7,8,9], thus corroborating the presence of a “thin red line” between an exasperated immune system activity and a neoplastic drift of the affected tissues.

Over the years, there has been an evolution of the concept of tumor microenvironment, initially considered as a tissue merely surrounding the tumor mass, indicating the cells composing the neoplastic niche participate actively in supporting the growth of carcinogen-altered cells to form focal lesions [10,11,12,13].

In the early stages of tumor development, the different immune cell populations, such as macrophages, neutrophils, mast cells, dendritic cells (DCs), and lymphocytes, intervene promptly with anti-tumor responses aimed at erasing the cancer cells [14]. Of note, in this context, it is conceivable that some neoplastic cell variants can acquire a less immunogenic phenotype thus escaping to immune detection [14]. Consequently, such immune-resistant selected clones start to release massively a plethora of chemotactic factors leading to the recruitment of immune cells within the neoplastic environment [15,16]. This is a crucial step in the neoplastic onset and development since a perverse partnership is created between cancer cells and the neo-infiltrating immune cells, through a paracrine and a cell-cell contact signaling, determining a phenotypical reorganization of immune cell population with a loss of their anti-tumorigenic functions [17,18]. In this context, the immune cells, releasing various cytokines and chemokines as well as oncogenic mediators (i.e., nitric oxide and growth factors), affect cell proliferation, death, and senescence, interfering also on DNA mutation rates and methylation, or the angiogenic process [19].

In this context, several authors provided interesting evidence about the involvement of adenosine, a retaliatory metabolite resulting from the degradation of adenosine triphosphate (ATP), in neoplasia onset and progression [20,21,22,23,24]. Under physiological conditions, low levels of adenosine are detectable in the interstitial fluids of unstressed tissues [25,26,27], whereas a marked increase of extracellular adenosine levels is observed under adverse conditions, including hypoxia, ischemia or inflammation [23,27,28]. Such an increase in the adenosine levels represents one of the pro-resolutive mechanisms aimed at suppressing and extinguishing an exuberant inflammatory reaction once its main task is attained [29]. It is worth to note that the prolonged persistence of high adenosine concentrations acquires detrimental features, triggering, and maintaining an immunosuppressed milieu, an ideal context for neoplasia onset and development [20,23,30].

Indeed, the marked presence of adenosine in the tumor microenvironment plays a critical role in shaping the generation of this niche, eliciting the repolarization toward an immunosuppressive phenotype for macrophages, DCs and neutrophils, with a contextual suppression of effector T cells and an expansion of regulatory T (T_reg_) cells [23,30]. In parallel, adenosine contributes to tumor growth, exerting a direct proliferative effect on neoplastic cells and sustaining either the neoangiogenic process and the extracellular matrix remodeling [31,32]. It is worth noting that such increased extracellular adenosine concentrations are the results of specific genetic alterations occurring during tumor progression [33]. Indeed, it has been well described that several tumors displayed an altered purine metabolism, characterized by a magnification of the molecular mechanisms facilitating the production of adenosine and by inhibition of the systems deputed to the degradation, thus creating an environment particularly rich in adenosine, suitable for cancer onset, development and spread [23,33].

The present review has been conceived to provide a comprehensive and critical overview about the involvement of adenosine system in shaping the molecular mechanisms underlying the enteric chronic inflammation and in promoting the generation of an immunosuppressive niche which supports the colorectal tumorigenesis.

## 2. Adenosine System: Enzymes, Transporters, and Receptors

Under physiological conditions, adenosine is present at low levels in both the intracellular and extracellular space [34]. In these conditions, the main source of intracellular adenosine is S-adenosylhomocysteine via S-adenosylhomocysteine hydrolase [35]. Extracellularly, adenosine results mainly by the degradation of ATP and adenosine diphosphate (ADP) to adenosine monophosphate (AMP), and then converted to adenosine, operated by the ectonucleotidases, CD39 and CD73, respectively [36] (Figure 1). The extracellular and intracellular levels of adenosine are finely tuned by the activity of the nucleoside transporters [29], classified into (1) equilibrative nucleoside transporters (ENT1, ENT2, ENT3, and ENT4), which transport nucleosides across cell membranes in either direction, based on concentration gradients [37,38,39] and (2) concentrative nucleoside transporters (CNT1, CNT2, and CNT3), which shunt extracellular adenosine into the intracellular space against their concentration gradient, exploiting the sodium ion gradient across cellular membranes as a source of energy [37,38,39]. Another critical checkpoint in the regulation of adenosine levels is represented by the expression and activity of adenosine deaminase (ADA), a key enzyme involved in the degradation of adenosine into inosine [35] (Figure 1).

Under pathological conditions, including hypoxia, ischemia, inflammation, or cancer, the concentration of adenosine increases rapidly as a consequence of massive extracellular dephosphorylation of ATP, mediated sequentially by CD39 and CD73 [40]. In this regard, several authors identified CD73 as a critical checkpoint in the regulation of extracellular adenosine levels and, consequently, in the control of receptor stimulation either under physiological or pathological conditions [41]. Of note, adenosine can also be generated through a non-canonical pathway by which nicotinamide adenine dinucleotide (NAD^+^)-glycohydrolase/CD38 enzyme axis converts extracellular NAD^+^ into adenosine diphosphate ribose (ADPR) [42]. ADPR is then metabolized by CD203a into AMP and then converted by CD73 into adenosine [42]. As described previously, also under adverse conditions the extracellular adenosine can be re-up taken into the cells through the nucleoside transporters, or converted into inosine by ADA [35,36,37].

Once released, the extracellular adenosine activates cellular signaling pathways through the engagement of the four known G-protein-coupled receptors, adenosine A_1_ receptor subtype (A_1_), A_2A_, A_2B_, and A_3_ [43]. A_1_ and A_3_ receptors are coupled with Gi, Gq, or Go proteins, whereas A_2A_ and A_2B_ are coupled to Gs or Gq proteins (Figure 1). The stimulation of A_1_ and A_3_ receptors can also trigger the release of calcium ions from intracellular stores, as well as A_2B_ receptor stimulation can also activate phospholipase C via Gq [44,45]. Of note, all the adenosine receptors are coupled to MAPK/ERK signaling pathways [44] (Figure 1). In addition, adenosine can also exert receptor-independent effects, via less defined intracellular mechanisms, including adenosine kinase, S-adenosylhomocysteine hydrolase systems, and AMP-activated protein kinase (AMPK) [46,47,48].

## 3. Adenosine System and Inflammatory Bowel Diseases (IBDs)

Inflammatory bowel diseases (IBDs), comprising mainly UC and CD, are chronic intestinal inflammatory disorders, characterized by excessive inflammation of the digestive tract, associated with thrombophilia and heightened risk of developing CRC [49]. Over the years, increasing evidence pointed out that the risk of cancer development in IBD patients is time-dependent and can increase by 2% by 10 years, 8% by 20 years, and 18% by 30 years [49,50]. The risk factors for CRC among IBD patients include severe inflammation, coexisting primary sclerosing cholangitis (PSC), family history of sporadic CRC, and age of colitis onset [49,50,51]. In particular, the risk of cancer in CD is controversial and, compared with UC, the risk is modest [52]. Based on these premises, research efforts have been focused on a better characterization of the molecular events determining a neoplastic degeneration of inflamed tissues in IBD patients, pointing out particular attention to the role exerted by adenosine system in the modulation of long-standing inflammation before the onset of tumorigenesis.

The majority of available pre-clinical studies have been performed using the dextran sulfate sodium (DSS)-induced colitis model, a widely used murine model, closely resembling the human UC, which allows a careful evaluation about the innate immune mechanisms involved in the development of bowel inflammation [53]. This model is based on the continuous administration in drinking water with DSS, a synthetic polymer of sulfated polysaccharides, leading to a reproducible acute inflammatory process, limited to the colon. Macroscopically, DSS administration is characterized by bloody diarrhea, intestinal inflammation and ulcerations, a marked body weight loss, and shortening of colonic length [53]. Histologically, this model is characterized by hyperosmotic damage toward the intestinal epithelial layer, which elicits a marked immune response in the host. In particular, the DSS colitis displayed a cytokine profile mainly characterized by a mixed T-helper type 2 (Th_2_) and Th_17_ immune paradigm [54].

Over the years, evidence has well described the role of adenosine as a potent endogenous immunoregulatory agent, able to blunt the exuberant inflammatory responses, shielding cells and tissues from an excessive immune response and immune-mediated damage [55,56]. Following an acute injury, a series of molecular mechanisms are put in place to increase the extracellular levels of adenosine, to elicit an efficient specific immune response aimed at restoring the tissue homeostasis [56]. In the late stage of inflammation, adenosine contributes relevantly to the resolution of inflammation, both repolarizing toward an anti-inflammatory phenotype of several innate immune cell populations (i.e., macrophages, DCs, neutrophils) as well as counteracting the acquired immune cell responses, acting directly on the T effector cells and spurring the expansion and the functions of immunosuppressive T_reg_ cells [56].

In this regard, studies performed on the pre-clinical DSS mouse model provided interesting evidence about the relevance of the adenosine system in orchestrating the immune responses (Figure 2). Siegmund et al. reported that an increase of endogenous adenosine levels, via pharmacological blockade of adenosine kinase by the selective inhibitor GP515, ameliorates DSS-induced colitis blunting the proinflammatory cytokine synthesis as well as suppressing interferon (IFN)-γ in colonic tissue [57]. Subsequently, Pallio and colleagues reported that adenosine, through A_2A_ receptor engagement, counteracted the expression of inflammatory cytokines, such as interleukin (IL)-1β and tumor necrosis factor (TNF), ameliorating significantly also the clinical features [58]. On the same line, Selmeczy et al. (2007) provided evidence about an ameliorative effect of CGS 21680, an A_2A_ receptor agonist, in the early stage of DSS-induced colitis [59] (Figure 2).

With regard to A_2B_ receptors, Kolachala et al. observed an over-expression of this receptor subtype in mice treated with DSS, demonstrating that such an increase was mediated by the activation of the TNF signaling pathway [60,61] (Figure 2). A relevant role of A_2B_ receptors in modulating the acute inflammatory phase of DSS colitis was described by Frick et al. (2009). Indeed the authors reported that the A_2B_ receptor engagement blunted the production of pro-inflammatory cytokines, such as IFN-γ, IL-1β, and IL-12, sparing an anti-inflammatory activity mediated mainly by IL-10 [62]. The administration of PSB1115, a selective A_2B_ receptor inhibitor, resulted in an increase in the severity of DSS colitis, thus corroborating a critical role of A_2B_ receptors in shaping the immune cell activity in a model of DSS-induced colitis [62]. In line with this view, Aherne et al. demonstrated that the genetic ablation of A_2B_ receptors was associated with increased severity of DSS colitis along with the loss of intestinal epithelial barrier functions [63]. Such alterations were mitigated by the treatment with selective A_2B_ receptor agonist, BAY-606583 [63] (Figure 2).

In the last years, increasing interest has been addressed to the pharmacological modulation of A_3_ receptors as a viable way to manage several immuno-mediated inflammatory diseases, including IBDs [29]. In this regard, it has been observed that the genetic ablation of the A_3_ receptor protected against DSS colitis effects [64]. By contrast, the pharmacological stimulation of A_3_ receptors with IB-MECA resulted effective in protecting against colitis through the reduction of inflammatory cytokine (i.e., IL-1β, IL-6, and IL-12) and chemokine (i.e., macrophage inflammatory protein (MIP)-1α and MIP-2) levels in colonic tissues [65] (Figure 2). On the same line, other authors highlighted the beneficial effect of 2-Cl-IB-MECA administration in DSS-treated mice, with a decrease of the pro-inflammatory responses through the inhibition of NF-kB signaling pathways [66]. Recently, it has been observed that the reduction of A_3_ receptor expression, via the treatment with miR-206-agomir (a microRNA analog), triggers the activation of pro-inflammatory NF-κB signaling pathways in DSS animal model [67], thus corroborating a “brake role” for A_3_ receptors on the inflammatory process. In support of this view, Ren et al. reported that the pharmacological activation of A_3_ receptors with 2-Cl-IB-MECA exerted an anti-inflammatory effect in human colonic epithelial cells, blunting the NF-κB signaling pathways along with a significant reduction of IL-8 and IL-1β production [68] (Figure 2).

Nowadays, some clinical studies have provided evidence about the anti-inflammatory effects of adenosine and its receptors in human colitis (Figure 2). For instance, a recent paper, by Tian and collaborators, reported that adenosine, through the activation of A_2A_ receptors, exerts a protective role in colonic inflammation in UC patients through the inhibition of NF-κB signaling pathway and the transcription of pro-inflammatory cytokine genes, including IFN-γ, IL-1β and IL-8 [69] (Figure 2). Other authors paid attention to the expression of A_2B_ receptors in human UC. Kolachala et al. observed an up-regulation of this receptor subtype in colonic tissues from a large cohort of UC patients, highlighting that A_2B_ receptor expression is modulated during inflammation [60,61] (Figure 2). With regard for the A_3_ receptor, Wu et al. observed that UC patients are characterized by an up-regulation of microRNA-206 (miR-206), which directly inhibits the expression of A_3_ receptors, with the consequent triggering of the pro-inflammatory response [67]. In line with this view, a more recent study confirmed the contribution of this receptor subtype in curbing the intestinal inflammation by inhibiting the production of TNF and IL-1β, along with an inhibition of NF-κB signaling pathway [70] (Figure 2).

Based on these premises, clinical and pre-clinical studies demonstrated that in the presence of intestinal inflammation adenosine represents a pro-resolutive mediator, acting as a modulator of the immune cellular responses to shield tissues from an excessive inflammatory response and immune-mediated damage. Indeed, adenosine, through the activation of specific receptors, inhibits the production of pro-inflammatory cytokines, including IFN-γ, IL-1β, IL-8, and IL-12, thus contributing relevantly to the resolution of inflammation. It is worth to note that the persistence of marked adenosine concentrations can become detrimental to tissues, triggering the secretion of immunosuppressive cytokines, such as IL-10 and transforming growth factor (TGF)-β from immune cells, overwhelming the immunosurveillance towards neoplastic cells.

## 4. Adenosine System and Immune Cell Interaction in Colitis-Associated Cancer Pathogenesis

Colitis-associated cancer (CAC) is a type of CRC preceded by clinically detectable IBD [71]. It is well known that long-standing enteric inflammation triggers chronic relapsing intestinal damage, thus creating a tumor-prone microenvironment [72]. In this context, adenosine exerts a critical role in tumor onset and progression, participating in the generation of an immunosuppressive and pro-angiogenic environment [23]. However, the molecular mechanisms through which the adenosine system participates in the various stages of transition from a chronic inflammatory context to the neoplastic transformation process remain poorly characterized. In this regard, the pre-clinical azoxymethane (AOM)/DSS mouse model represents a useful tool to deepen the mechanisms underlying the onset of CAC to define novel therapeutic approaches [73]. Indeed, the murine model combining the AOM, a colonic genotoxic carcinogen, with DSS, a colitis inducer, shortens the latency time for induction of CRC leading to the aberrant crypt foci-adenoma-carcinoma sequence that occurs in human CAC [73].

During cancer pathogenesis, the extracellular levels of adenosine increase significantly, reaching the micromolar range [74,75]. In this regard, an in situ microdialysis study performed in colonic specimens from human and mouse, reported a 20-fold increase of extracellular adenosine when compared to normal tissues [20]. The persistence of increased adenosine concentrations beyond its anti-inflammatory and pro-resolutive task, becomes detrimental for the tissues, generating an immunosuppressed milieu useful for the onset and development of neoplastic cells [30]. Several investigations highlighted that such extracellular adenosine accumulation results from specific genetic alterations occurring during the carcinogenic process [33].

Increasing evidence demonstrates an increase in the expression and/or activity of CD39 and CD73 in tumor endothelial cells as well as in immune cells (i.e., T_reg_ cells, myeloid-derived suppressor cells (MDSCs), macrophages, DCs, and Th_17_ cells) isolated from patients with CRC [76,77,78,79,80]. Such increased expression and activity of the CD39-CD73 axis leads to a chronic presence of high adenosine levels in the tumor microenvironment (TME), impairing markedly the anti-tumor immune response and thus paving the way to cancer cell proliferation and angiogenic process. For this reason, the concomitant inhibition of CD39 and CD73 activity could represent an innovative pharmacological strategy in cancer therapy, aimed at counteracting tumor growth by awakening the anti-tumor immunity. Of note, the prolonged and persistent condition of inflammation and hypoxia, commonly observed in CAC patients, are considered the most frequent inducers of CD39 and CD73 expression and activity [81]. In this regard, several studies have reported that the hypoxia-inducible factor (HIF)-1α triggers an increase of CD39 and CD73 activity, thus impairing the T cell functions via the generation of high concentrations of adenosine [82,83,84,85,86,87].

In parallel, it has been observed a down-regulation of the catabolic enzyme ADA and its cofactor CD26 (also known as dipeptidyl peptidase 4) [88,89,90] as well as the reduction of adenosine kinase activity [91] in the cancer microenvironment.

Once released in TME, extracellular adenosine exerts a potent immunosuppressive and cancer growth-promoting effect interacting with several immune cell populations [21]. In particular, adenosine, beyond to counteract the effector T cell functions mainly via the engagement of the A_2A_ receptors, inhibits mononuclear phagocyte cell differentiation and maturation as well as contributes to the angiogenetic processes and matrix remodeling environment, suitable for cancer growth [92,93,94,95]. Of note, the effect of adenosine on the immune system depends on its bioavailability and receptors engaged in immune cells present within TME. For instance, the activations of A_2A_ receptors spur the T_reg_ functions, via cAMP/protein kinase cAMP-dependent (PKA) pathways, thus promoting immune-suppressive effects [96,97,98,99,100] (Figure 3). In support of this view, other studies reported also that A_2A_ receptor activation prevents natural killer (NK) cell activation, maturation, and cytotoxicity [92,99].

The marked presence of adenosine in the TME, engaging the A_2A_ receptors expressed on macrophages and DCs, stimulates the IL-10 release and counteracts the IL-12 production, with subsequent impairment of T cell priming and the suppression of antitumor immune responses [96,100,101] (Figure 3). Of note, in several murine cancer models, including the CAC model, it has been observed a potent antitumor effect exerted by IL-12, via the reduction of distant metastases formation and prolonging significantly the survival of mice allowing an efficient anti-tumor response [102,103].

Increasing attention has been paid to the role of A_2B_ receptor in controlling immune cell-mediated responses. In this regard, Ryzhov et al. reported that the activation of this receptor subtype promoted the expansion of MDSCs, an immature subset of myeloid cells, which express high levels of CD73, thus contributing to a further generation of adenosine with consequent increase of the immunosuppressive environment [104] (Figure 3). In parallel, it has been observed that the engagement of A_2B_ receptors contributes to the generation of an immunosuppressive milieu, stimulating the T_reg_ differentiation and proliferation along with an increase in IL-10 production [105,106]. In the TME, the engagement of the A_2B_ receptors expressed on macrophages, elicits the secretion of vascular endothelial growth factor (VEGF) from these cells thus playing a relevant role in tumor angiogenesis [106] (Figure 3). In support of this view, the pharmacological blockade of A_2B_ receptors with the selective antagonist PSB1115, reduces the release of VEGF and the number of tumor-infiltrating MDSCs in a mouse melanoma model, thus corroborating the pro-angiogenic and immunosuppressive effects of this receptor subtype [106].

This evidence was corroborated by further studies reporting that the extracellular adenosine, via A_2A_ and A_2B_ receptor engagement, counteracted the neutrophil activation as well as their adhesion to the endothelial cells, participating to the generation of an immunosuppressive and proangiogenic milieu [104,107,108].

Interestingly, Harish and colleagues demonstrated that the activation of A_3_ receptors expressed on NK cells increases their antitumor activity [109]. Recently, it has been also observed that the engagement of A_1_ and A_3_ receptors on neutrophils mediate multiple immunosuppressive effects, including the inhibition of oxidative burst response [110] (Figure 3).

At present, clinical evidence about the role of the adenosine system in the molecular mechanisms underlying the inflammation-associated CRC pathogenesis are scanty. An over-expression of CD39 and CD73 has been reported in T_reg_ and MDSCs isolated from CRC patients [77,111], suggesting that tumor-infiltrating T_reg_ and MDSCs are likely an important source of extracellular adenosine that contributes to the tumor immune escape. In the same study emerged that adenosine resulting from circulating human CD39^+^ T_reg_ and MDSCs, beyond stimulating the vascular endothelial cell proliferation, reduced markedly the migration of effector T cells into the TME, thus disabling the anti-tumor immunity [77,111].

An over-expression of A_2B_ receptors was also observed in sections of neoplastic colorectal tissues in comparison with normal colonic tissues [112]. In the same study, the authors detected a marked presence of this receptor subtype in several human CRC-derived cell lines [112]. In this context, the treatment with selective A_2B_ receptor antagonist, MRS1754, reduced the neoplastic growth of CRC cell lines, indicating a direct cancer-promoting property of this receptor subtype [112]. In this regard, other authors reported that also the A_3_ receptors, overexpressed in CRC tissues, can take a relevant part to CRC proliferation [113].

In summary, the extracellular adenosine, massively released in cancer tissue, participates actively to the generation of immunosuppressive and cancer growth-promoting microenvironment by regulating the functions of tumor cells, stromal cells, and tumor-infiltrating immune cells. In this regard, adenosine, through the engagement of its receptors expressed on immune cells, switches the phenotype of these cells from immune surveillance and host defense role towards a cancer-promoting phenotype, contributing to the generation of immunosuppressed niche. At the same time, beyond its involvement in the generation of an immune tolerant microenvironment suitable for tumor onset and progression, adenosine interferes also directly with cancer cells, spurring their proliferation, inhibiting the apoptotic process, promoting the angiogenic process, and predisposing the neoplastic cells to the metastatic dissemination.

## 5. Conclusions

Clinical and pre-clinical studies indicate that adenosine acts as a potent endogenous anti-inflammatory and immunoregulatory agent, able to shape the phenotype and the activity of immune-inflammatory cells. However, the excessive depression of the immune system induced by the marked presence of adenosine can exacerbate tissue dysfunction in chronic diseases, hindering the anti-tumor immunity and thus promoting the tumor progression.

In UC, the increment of extracellular adenosine levels, resulting from an increase in CD39 and CD73 expression and activity [76,114,115], represents one of the pro-resolutive mechanisms aimed at attenuating tissue inflammation and immune-mediated damage (Figure 4). In this context, adenosine, through the engagement of specific receptors expressed on immune cells, exerts an immunosuppressive activity curbing the pro-inflammatory cytokine synthesis, including IL-1β, IL-8, and IL-12, thus exerting a beneficial effect on intestinal inflammation. However, a marked and prolonged presence of high levels of adenosine could promote cancer initiation and progression; therefore, the pharmacological modulation of the molecular mechanisms aimed at producing adenosine (i.e., CD39-CD73 axis) as well as the blockade of adenosine receptors actively involved in switch off the immune system (i.e., A_2A_ and A_2B_) represents an attractive strategy to counteract the onset of tumorigenesis [116].

Interestingly, in inflamed areas of cancer tissue, it has been observed an increase CD39 and CD73 expression and activity along with a reduction in ADA and CD26 expression, resulting in a marked and persistent increase in extracellular adenosine concentrations (Figure 4). Such accumulation of adenosine within the intratumoral milieu induces a persistent secretion of immunosuppressive cytokines, such as IL-10 and TGF-β, contributing to tumor immune evasion.

An interesting point about the current evidence from the literature is the role played by exosomes released from cancer and stromal cells within TME [117,118,119]. The exosomes are vesicles with a diameter ranging from 30–100 nm, characterized by the expression of CD39 and CD73 on their surface exhibiting a marked ATP-AMP phosphohydrolytic activity [117]. Adenosine deriving from exosomes displayed to exert an inhibitory effect on T cell activity as well as to support the expansion of inhibitory regulators of the immune system, such as MDSCs, thus contributing to the generation of immunosuppressed milieu (Figure 4). However, despite this intriguing evidence, there are still many dull aspects that deserve further investigations. In particular, among the questions still open (1) what are the effects of adenosine on mast cells in the TME? This issue is not well described and deserves more focused studies; (2) what about the molecular mechanisms underlying the adenosine-mediated immunosuppression in TME of CAC? (3) what a role for adenosine released from exosomes in the onset and progression of CAC? (4) what are the role and the impact of adenosine in the pathogenesis of CAC? To address these points, intensive research efforts on AOM/DSS mouse model should be implemented to better clarify such molecular events.

A better characterization about the complex molecular mechanisms driving the interplay between adenosine system and immune cells could pave the way to the identification of novel key factors underlying these interactions, thus prompting the development of “checkpoint blockades” able to counteract these immunosuppressive effects in the TME. Based on the above-mentioned evidence, it appears that the pharmacological modulation of the adenosine system represents an attractive strategy to improve the clinical response to other immunotherapy as well as chemotherapy and radiotherapy [120]. In line with this view, several clinical trials are currently in progress to evaluate the putative beneficial effect of pharmacological tools acting on adenosine system alone or in combination in the management of neoplastic disorders [120]. For instance, clinical trials are underway for AB928, a dual A_2A_/A_2B_ receptor antagonist, combined with chemotherapy in CRC patients [121]. Other clinical trials in phase 1/2a are underway to assess the safety and efficacy of a monoclonal antibody that inhibit CD73, MEDI9447, combined with chemotherapy and radiotherapy in patients with advanced solid tumors, including CRC [121]. At present, the development of inhibitors of CD39 for cancer therapy is underway, but none have yet entered the clinic.

## Figures and Tables

**Figure 1 ijms-21-05089-f001:**
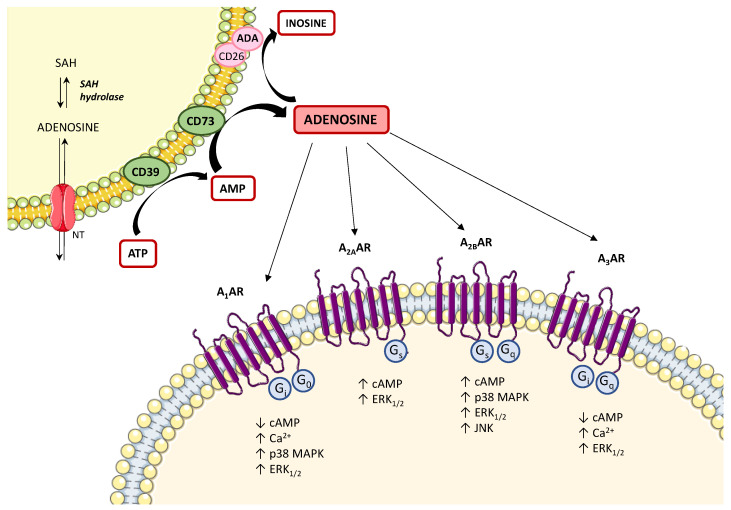
Diagram showing the main step of adenosine biosynthesis and catabolism and the second messenger pathways coupled to adenosine receptors. The main source of intracellular adenosine is S-adenosylhomocysteine (SAH) via S-adenosylhomocysteine hydrolase (SAH hydrolase) and then released into the extracellular space via nucleoside transporters (NTs). The extracellular adenosine is mainly produced by the ectonucleotidases, CD39 and CD73. Excess adenosine is irreversibly deaminated to inosine by the enzyme adenosine deaminase (ADA). In the extracellular space, adenosine can bind to four different G protein-coupled adenosine receptors that either inhibit (mediated by A_1_ and A_3_ adenosine receptors) or stimulate (mediated by A_2A_ and A_2B_ adenosine receptors) adenylyl cyclase activity and cAMP production in the cell. All adenosine receptors couple to MAPK pathways, including ERK_1/2_ and p38 MAPK. ↑: increase; ↓: decrease; A_1_AR: A_1_ adenosine receptor; A_2A_AR: A_2A_ adenosine receptor; A_2B_AR: A_2B_ adenosine receptor; A_3_AR: A_3_ adenosine receptor; ATP: adenosine triphosphate; Ca^2+^: calcium ions; cAMP: cyclic AMP; ERK_1/2_: extracellular signal-regulated kinase 1/2; JNK: JUN N-terminal kinase; MAPK: mitogen-activated protein kinase.

**Figure 2 ijms-21-05089-f002:**
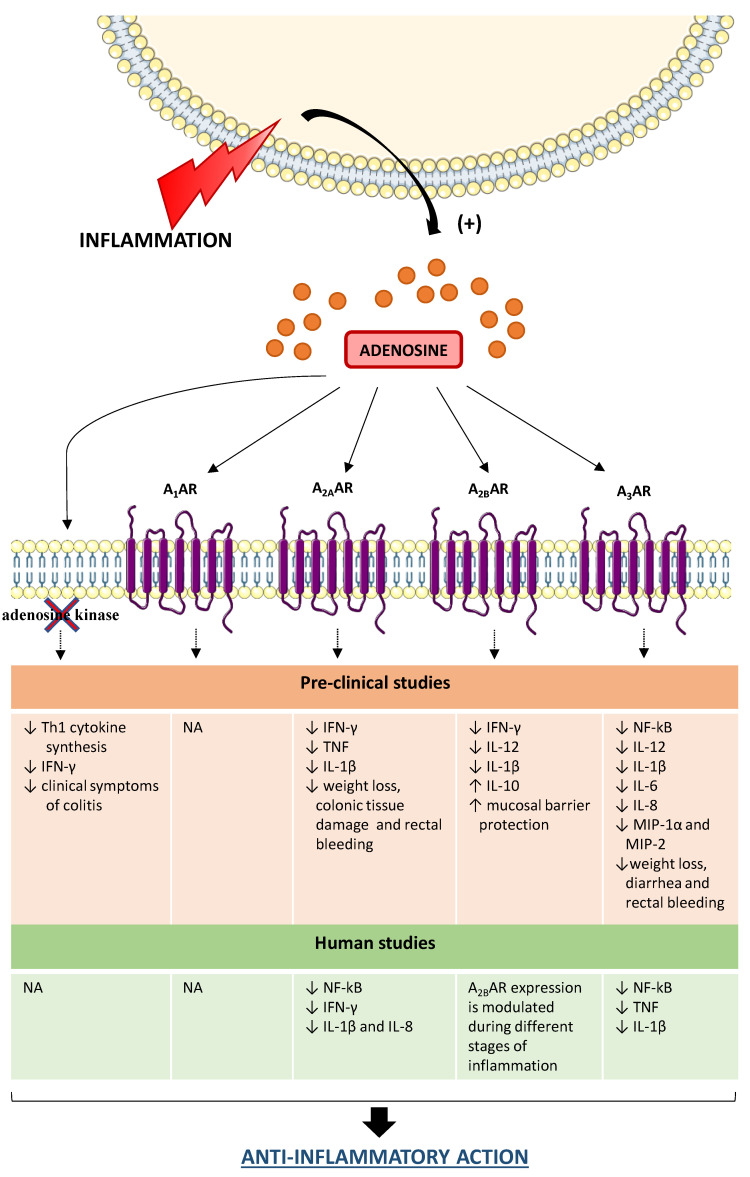
Role of adenosine receptors in the modulation of the inflammatory response in IBDs. In the presence of intestinal inflammation, the extracellular levels of adenosine increase significantly to restore tissue homeostasis. In this context, adenosine exerts an anti-inflammatory action, thus shielding cells and tissues from an excessive inflammatory response and immune-mediated damage. ↑: increase; ↓: decrease; A_1_AR: A_1_ adenosine receptor; A_2A_AR: A_2A_ adenosine receptor; A_2B_AR: A_2B_ adenosine receptor; A_3_AR: A_3_ adenosine receptor; IL: interleukin; INF- γ: interferon-γ; MIP: macrophage inflammatory protein; NA: not available; Th1: T-helper type 1; TNF: tumor necrosis factor.

**Figure 3 ijms-21-05089-f003:**
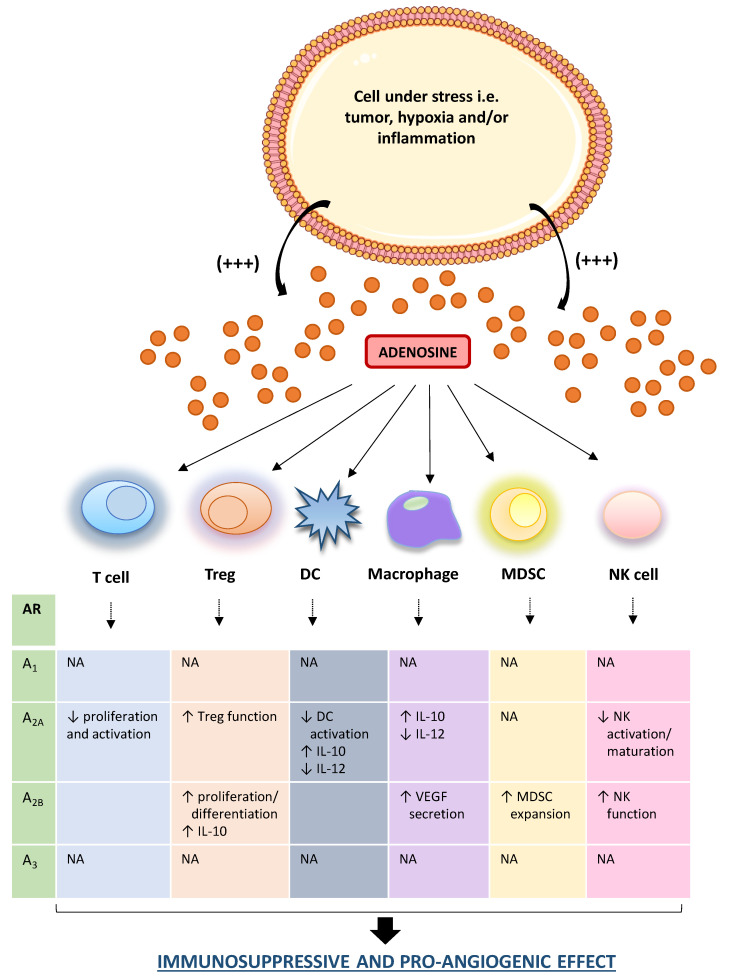
Schematic figure of adenosine and immune cell interactions in TME. In the tumor microenvironment, adenosine is markedly released in the extracellular space. In this context, it interacts with infiltrating immune cells (i.e., macrophages, T cell, DCs), triggering, and maintaining an immunosuppressed milieu, an ideal condition for the onset, and development of neoplasia. ↑: increase; ↓: decrease; AR: adenosine receptor; DC: dendritic cell; IL: interleukin; MDSC: myeloid-derived suppressor cell; NA: not available; NK: natural killer.

**Figure 4 ijms-21-05089-f004:**
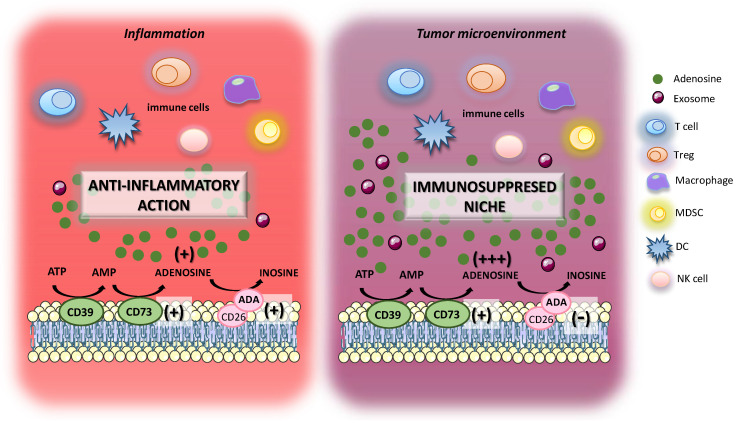
Role of adenosine system in enteric inflammation and intestinal tumor microenvironment. In the presence of enteric inflammation, the extracellular levels of adenosine increase significantly to restore tissue homeostasis. In this context, it has been observed an over-expression of CD73 along with an increase in ADA expression levels. By contrast, in TME it has been reported an increase in CD73 expression and activity, along with a down-regulation of ADA and its cofactor CD26 with consequent accumulation of adenosine within the intratumoral milieu, resulting in an immunosuppressed niche.

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
