# Peer review of "The Adenosine System at the Crossroads of Intestinal Inflammation and Neoplasia"

_ijms, 2020, doi:10.3390/ijms21145089_

Round 1

Reviewer 1 Report

The present review provides a comprehensive and critical overview about the involvement of adenosine system and the molecular mechanisms associated to enteric chronic inflammation and colorectal tumorigenesis. The manuscript is well-written and characterizes the complex molecular mechanisms driving the interplay between adenosine system and immune cells suggesting that adenosine system could represent a new strategy to improve the clinical response to the treatment of the neoplastic disorders.

Author Response

We wish to thank the Reviewer for comments concerning the review article

Reviewer 2 Report

The authors D’Antongiovanni et al. have provided a comprehensive and critical overview of the involvement of the adenosine system in the modulation of intestinal inflammation and colorectal tumorigenesis. The review is clear, scientifically sound and interesting. Figures are very helpful for the reader and highly increase the value of the manuscript.

Minor points:

  1. 2 L.75 Treg should be defined
  2. 2 L.90 “in this conditions” should be “in these conditions”
  3. 4 L.153 “DSS-administration” should be “ DSS administration”
  4. 5 L.158 Th should be defined
  5. 5 L.177 Please describe what CGS 21680 is
  6. 6 L.222 “pro-resolutive mediators” should be “pro-resolutive mediator”
  7. 8 L.239 “immune cells interactions” should be “immune cell interaction”
  8. 8 L.264 TME should be defined
  9. Figure 3. Typo: Activatio/maturation
  10. 10 L.348 The sentence “However, the marked depression exerted by the marked presence of adenosine on the immune system, can lead to tissue dysfunctions” is a bit awkward. I would suggest to rephrase it, but this is up to the authors.

Reviewer 3 Report

D'Antogiovanni et al. present a nicely written review about the role of adenosine and it's receptors in the context of IBD and CAC. I personally like the systematic Arrangement (mouse -->human; IBD -->CAC). Especially the figures are very good and alsmost selfexplaining. Hence, I consider it worth publishing.

Unfortunately, there are a few minor aspects making the review worse than it is actuallay.

1) The language and style could be a little bit better. There are a few Errors (e.g. line 260 "well demonstrated" --> demonstrated well would be correct. There are more of this minor errors through out the manuscript.

Another aspect is the overuse of the words "niche" and "elicit" making it boring to read these words repeatedly.

2) There are too many reviews in the reference list. More orginal works (in the first refferences) would be better.

3) The basics of Tumor development should be clearified: more CU than MC (the sentence in line 143 sounds strange), PSC as a trigger, duration and amount of inflammation, onset age.

4) I really like the ending of the review revealing open question. Still, I would wish some stamtents about possible therapeutic approaches for humans.  
